# Differences in PCV13 Recommendation Practices between Pediatric Care Providers and Primary Care Providers in China: A Cross-Sectional Survey of Behavior and Social Drivers

**DOI:** 10.3390/vaccines12091082

**Published:** 2024-09-22

**Authors:** Yuan Dang, Lin Wang, Yuming Liu, Boyan Wang, Huiwen Deng, Can Ye, Chunping Wang, Yangmu Huang

**Affiliations:** 1School of Public Health, Peking University, Beijing 100191, Chinakimliu082@gmail.com (Y.L.); 2210306142@stu.pku.edu.cn (B.W.); 2210306131@stu.pku.edu.cn (H.D.); 2210306145@stu.pku.edu.cn (C.Y.); 2Child Health Care Center, Children’s Hospital, Capital Institute of Pediatrics, Beijing 100020, China; 3School of Public Health, Weifang Medical University, Weifang 261053, China; chpwang@163.com

**Keywords:** pediatric care providers, primary care providers, behavior and social drivers, pneumococcal vaccination, recommendation practice, cross-sectional survey

## Abstract

**Objectives:** This study examines the recommendation behaviors and influencing factors for the 13-valent Pneumococcal Conjugate Vaccine (PCV13) among 3579 Chinese healthcare workers (HCWs), including 1775 pediatric care providers (Peds-PCPs) and 1804 primary care providers (PCPs). Data were collected from May to July 2023 through a national cross-sectional survey using a structured questionnaire, distributed across hospitals providing pediatric services in five provincial-level administrative divisions. **Methods:** The sociodemographic data, vaccine knowledge, and recommendation practices were analyzed using Pearson’s chi-square test, Wilcoxson rank-sum test, and multivariate logistic regression. Results show that while PCPs are more likely to recommend PCV13, vaccine hesitancy persists among Peds-PCPs. Logistic regression revealed that higher influenza vaccination intention, salary, vaccine consultation frequency, familiarity with immunization, work ethic, and flexible schedules positively impacted HCWs’ recommendation behavior. **Results:** Factors influencing Peds-PCPs’ recommendations include vaccine training (OR: 1.470, CI: 1.049–2.509), safety recognition (OR: 1.986, CI: 1.163–3.391), concern over rejection (OR = 1.274, CI: 1.076–1.508) and vaccine cost (OR = 1.203, CI: 1.023–1.414). For PCPs, influencing factors were the perceived susceptibility of children to pneumonia (OR = 2.185, CI: 1.074–4.445), acceptance of herd immunity (OR: 1.717, CI: 1.101–2.677), and belief that parents with better family conditions are more likely to accept vaccine recommendations (OR = 1.229, CI: 1.024–1.477).  **Conclusion:** This survey underscores the need for tailored interventions to address differing perceptions and enhance confidence in the safety and efficacy of vaccines among HCWs, particularly Peds-PCPs.

## 1. Introduction

Mucosal diseases and invasive pneumococcal disease (IPD) caused by Streptococcus pneumoniae have a high morbidity and mortality rate worldwide [1]. Pediatric populations under 2 years old are more vulnerable to pneumococcal infection. There are approximately 12% of total cases worldwide occurring in China [2], and an estimated 218,200 severe IPD cases with 8000 IPD deaths occurred in Chinese children under 5 years old in 2017 [3]. At present, the pneumococcal conjugate vaccine (PCV) has proven to be highly effective in reducing associated hospitalizations and related mortality. The WHO published a position paper to recommend all countries include pneumococcal conjugate vaccine (PCV) in their national immunization program (NIP) as early as 2007 [4]. Among the 194 WHO member states, 148 countries had introduced PCV into their routine infant immunization schedule at the end of 2020 [5].

China’s national immunization program, funded by the central government, has been expanded several times since its establishment, which provides free vaccination to all entitled children for 14 vaccines covered under the program. Unfortunately, the 13-valent PCV (PCV13) has yet to be included in China’s NIP, making it a completely self-pay vaccine with no subsidies available. A variety of factors, such as disease burden, vaccine availability, vaccination capacity, cost-effectiveness, need to be considered when including a vaccine in the NIP. Consequently, the pooled vaccination rate of PCV13 in China is only 21.7% (95% CI: 17.2–26.5%), substantially lower than the global average [6] and that of most Asian countries as reported by the WHO immunization data [7]. Within the Western Pacific Region of the WHO, the PCV13 coverage is 26%, while the European Region boasts significantly higher vaccination rates of 86% for PCV13, underscoring a regional disparity. The gap between NIP vaccine coverage and PCV13 coverage highlights a significant area for improvement.

As drivers of parents’ vaccination decisions are multifaceted, vaccine hesitancy among healthcare workers (HCWs) plays an important role in the parents’ vaccination perceptions and decision-making [8]. There have been increasing reports of vaccine hesitancy among healthcare workers in European countries, highlighting how their attitude can significantly influence parents’ decisions [9]. This could be particularly relevant in China, as immunization information from HCWs in local hospitals and primary healthcare facilities is often considered the most trustworthy by many parents [10]. However, previous studies have shown that 46.2% of public health workers in China were not willing to recommend any non-NIP vaccines to children, which is a barrier to optimizing non-NIP vaccine uptake [11]. A critical element that contributes to vaccine hesitancy is trust. Trust between providers and caregivers form a cycle where providers exhibit confidence in vaccines and communicate this effectively; this fosters trust among caregivers, which in turn encourages them to accept vaccination for their children. Conversely, if HCWs themselves express distrust or hesitancy regarding vaccines, it undermines caregiver trust, resulting in a counterproductive feedback loop that impedes vaccination efforts. Healthcare workers are not required to recommend non-NIP vaccines as part of their official duties; such recommendations are voluntary. However, the willingness of different types of healthcare workers in China to recommend PCV13 remains unclear.

Primary care providers in China deliver a range of preventive services including vaccinations, health education, and medical services such as treating common diseases. Pediatric care providers, on the other hand, work in children’s hospitals or general hospitals, often dealing with children who have medical conditions. Therefore, Ped-PCPs, who are widely recognized by the children’s parents for their professional expertise, often garner a profound level of trust from these parents, despite not being directly responsible for administering vaccination services. Until now, few studies have been published on the recommendation practice for PCV13 of both PCPs and Ped-PCPs. Thus, the recommendations from both PCPs and Ped-PCPs are important for shaping the immunization decisions of parents.

In 2022, the SAGE of the WHO released a position paper on using the behavior and social drivers (BeSD) tools of vaccination with four domains: thinking and feeling, social processes, motivation, and practical issues, which influence the decision-making process during vaccination-related behavior [12]. Therefore, this study attempted to describe and compare the drivers and barriers of PCV13 recommended practice for both Ped-PCPs and PCPs following the BeSD framework. Recent systematic review has highlighted a gap in research on vaccine hesitancy and trust in vaccination from the perspective of healthcare providers, particularly in low- and middle-income countries [13]. Our research aims to address this gap by identifying factors that affect the providers’ vaccine recommendations. Specifically, this study identified changeable factors across four BeSD domains that hinder the recommendation by healthcare providers for PCV13. By understanding these barriers, we can design and promote more targeted interventions to change provider behavior and improve PCV13 coverage in a wider population.

## 2. Materials and Methods

### 2.1. Study Design and Participants

For this observational, cross-sectional survey, data were collected from 10 May 2023 to 29 July 2023, across 78 medical institutions in five representative provincial-level administrative divisions (PLADs) in China, reflecting a range of socio-economic development levels. With informed consent obtained, the target population of the survey was divided into two groups. Inclusion criteria: (1) licensed pediatric care provider, which included specialist medical doctors and nurses working in pediatric departments, respiratory departments, or pediatric intensive care units of children’s hospitals or large-scale medical centers, and (2) primary health workers (consisting of general practitioners, nurse, public health workers) holding a vaccination certificate working at community healthcare clinics to provide vaccine-related services. Exclusion criteria: healthcare workers who did not agree to participate. To confirm the eligibility of then individual participants, each subspecialty department’s liaison was asked to forward the survey to all of their team members so that we could collect an unbiased sample.

The study team developed a structured questionnaire regarding sociodemographic, vaccine knowledge, and recommendation-related information and performed the survey via Wenjuanxing (a widely accepted online platform in China for data collection). On the premise of voluntary participation in the study, all data were de-identified and the participants were anonymous. It took about 10 min to complete the questionnaire, and each participant could receive a RMB 20 (equivalent to USD 2.80) allowance for their time and support upon answering all questions. The study protocol was submitted to the Survey and Behavioral Research Ethics Committee of Peking University (IRB00001052-23030) and approved before the data collection.

The initial sample size n required for the survey was calculated by the following formula based on an error of 0.05 and a maximum permissible error
$\delta :n=\frac{z_{1-\alpha /2}^{2}\cdot p\cdot (1-p)}{\delta^{2}},\delta =0.1p$. *p* represents the predicted PCV13 recommendation rate of HCWs, referring to 50% (the most conservative estimated sample size). We calculated that the minimum total sample size of each group of HCWs was expected to be 1095, which is to say, 219 for the two types of HCWs in each municipality. To allow for the disqualification of incomplete questionnaires, we increased the sample size by 10%, with a final target sample population of 1205 for each group. In practice, we collected a larger sample size than expected to increase the reliability including 1775 Ped-PCPs and 1804 PCPs. In this survey, we adopted a multistage sampling method. First, five PALDs (Chongqing, Jilin, Shandong, Shenzhen, Beijing) were selected based on China’s Division of Central and Local Financial Governance and Expenditure Responsibilities in the Healthcare Sector [14], which stratifies the 31 PALDs into five layers. Second, within each province or municipality, medical institutions were classified into three levels based on the region’s economic development level (GDP): low, medium, and high. We ensured that the minimum sample size across these three levels was satisfied, with at least 73 participants from each location for the PCP group and Peds-PCP group.

### 2.2. Measures

The self-administered questionnaire for healthcare workers was designed based on the behavioral and social drivers of the vaccination framework and previous studies on the attitudes and behavior of HCWs toward vaccine recommendations [15,16]. The guidelines published by the Department of Immunization, Vaccines and Biologicals (IVB), WHO [17] provided the knowledge basis for the questionnaire. The preliminary questionnaire underwent testing via face-to-face interviews with pre-selected participants. Initially, the respondents completed the questionnaires independently. Subsequently, they were interviewed about their comprehension of the questions and any challenges encountered during completion. Based on their feedback, adjustments were made to content and phrasing, finalizing the questionnaires. The questionnaire collected demographic information and socioeconomic status including gender, age, education level, professional title, work time, average annual basic salary, etc. Additionally, information regarding the recommendation behaviors of PCV13 and the reasons for not recommending PCV13 were also obtained. Knowledge assessment comprised both knowledge about pneumococcal disease and PCV13 for which the participants responded with “True” or “False”. We used a five-point Likert scale for survey items that assessed attitude toward the PCV13 vaccine including thinking and feeling, motivation, social process, and practical issues. These questions were close-ended with possible responses including “strongly disagree”, “disagree”, “neutral, agree”, and “strongly agree”.

### 2.3. Statistical Analysis

All categorial and Likert-scale variables were presented as frequencies and proportions. To test the difference between Ped-PCPs and PCPs, Pearson’s chi-square test and the Wilcoxon rank-sum test were conducted for comparing categorical variables and Likert-scale variables, respectively. Bar charts were used to visually present the data. For univariate analysis, the chi-square test and Wilcoxon rank-sum test were performed to assess the association between the PCV13 recommendations with each of the independent variables. Independent variables tested to be significant were then further entered into the multivariate logistic regression model where odds ratios (OR) and 95% confidence intervals (CI) were calculated to examine the influencing factors of the recommendation behaviors of healthcare workers. Variance inflation factors (VIFs) and tolerances were used to investigate for variable collinearity, with a VIF less than 5 and tolerance greater than 0.1 considered to indicate no significant collinearity. The accuracy and stability of the models were evaluated using the Hosmer–Lemeshow goodness-of-fit test. Two-sided *p*-values < 0.05 indicated statistical significance. All data were analyzed with SPSS version 26.0 (Armonk, NY, USA) and PyCharm 2023.3.2 (Community Edition).

## 3. Results

### 3.1. Study Sample Characteristics

Table 1 presents the characteristics and vaccine-related attitudes of 1775 Ped-PCPs and 1804 PCPs.

For the Peds-PCPs, 71.72% were female, with a mean age of 35.83 years (SD 9.48) and 1.86 years (SD 5.92) of vaccination experience. A total of 11.10% (n = 1775) had a master’s degree or higher, and worked an average of 8.27 h per day, with most being doctors (45.30%), and 36.62% holding intermediate professional titles. A total of 75.21% received vaccine-related education during their studies, and 74.03% updated their knowledge in the past two years. A total of 70.87% was dissatisfied with their salary.

For PCPs, 83.09% (n = 1804) were female, with a mean age of 37.35 years (SD 9.03) and 6.53 years (SD 7.57) years of vaccination experience, 4.43% (n = 1804) had a master’s degree or higher, worked an average of 7.98 h per day, with more nurses than doctors (47.39% vs. 29.21%). A total of 38.47% held professional or junior titles; 74.22% received vaccine-related education during their studies; and 91.13% updated their knowledge in the past two years. A total of 79.10% was dissatisfied with their income.

### 3.2. Attitudes and Behaviors for Recommending PCV13

Table 2 displays the difference in PCV13 recommendation between the two types of HCWs. The overall PCV13 recommendation rate of pediatric care providers was 41.02%, and that of the primary care providers was 68.79%. Particularly, 1.88% of primary care providers and 10.76% of pediatric care providers never recommended PCV13 vaccination to children. The means of the frequency scores of Peds-PCP and PCPs in recommending the PCV13 vaccine differed, with the mean score for PCP being slightly higher at 3.89 ± 0.023. The difference was significant (*p* < 0.001) both in the chi-square test and the Wilcoxon rank-sum test.

For healthcare workers, the possible occasions to introduce information about PCV13 or recommend the vaccine included health education courses, outpatient service, and routine vaccination. It was found that the proportions of HCWs in primary hospitals or community health centers who had recommended PCV1 during health education courses, outpatient service, and routine vaccination were 72.86%, 71.55%, 68.18%, respectively, and these proportions were higher than the overall recommendation rate (69.90%, 71.01%, 69.68%, respectively). Compared with other scenarios, there was a higher proportion of HCWs who were more likely to introduce PCV13 when the parents actively consulted about pneumonia-related information (Table 3).

### 3.3. Influencing Factors

In our study, the recommendation behaviors of HCWs were quantified as to whether to recommend the PCV13 vaccine to parents. The collinearity diagnostic analysis showed that the VIFs of those risk factors were less than 4, suggesting that there was no strong indication of multicollinearity among variables (Appendix A). The results of the Hosmer–Lemeshow test indicated that both the Ped-PCP (*χ*^2^ = 9.569, *p* = 0.297) model and the PCP model (*χ*^2^ = 6.492, *p* = 0.592) exhibited good overall fits. Table 4 shows the results of two individual multiple logistic regressions adjusted for sociodemographic characteristics to identify the influencing factors of PCV13 recommendation among the HCWs. Generally, Ped-PCPs (OR = 1.245, CI: 1.021–1.518) and PCPs (OR = 1.649, CI: 1.146–2.373) with higher average basic salary, Ped-PCPs (OR = 1.238, CI: 1.050–1.458) and PCPs (OR = 1.550, CI: 1.180–2.036) who were willing to get the influenza vaccine in the future had higher odds of recommending the PCV13 vaccine. Meanwhile, Ped-PCPs (OR = 4.544, CI: 3.721–5.547) and PCPs (OR = 2.338, CI: 1.850–2.956) who had frequent vaccine consultation for parents, Ped-PCPs (OR = 1.277, CI: 1.072–1.522) and PCPs (OR = 1.692, CI: 1.340–2.137) who were familiar with PCV13 vaccination procedures, and Ped-PCPs (OR = 1.353, CI: 1.167–1.567) and PCPs (OR = 1.442, CI: 1.193–1.744) who agreed that it was their responsibility to provide information and advice on PCV13 vaccines to parents of children were significantly associated with PCV13 vaccine recommendation. Furthermore, some Ped-PCPs (OR = 0.750, CI: 0.637–0.882) and PCPs (OR = 0.645, CI: 0.514–0.810) agreed that their busy schedules prevented them from introducing the pneumonia vaccine to parents. 

For Ped-PCPs, we found that workers who had vaccine training history in the past two years (OR = 1.470, CI: 1.049–2.509), known that the newly marketed PCV13 did not carry a higher risk than PCV7 (OR = 1.986, CI: 1.163–3.391), and perceived a high necessity of vaccination (OR = 1.635, CI: 1.115–2.397) thought that recommending PCV13 to parents would be undesirable (OR = 0.785, CI: 0.663–0.929). The main reason why parents were reluctant to vaccinate their children against pneumonia was the high price (OR = 0.831, CI: 0.707–0.977), which was positively related to the Ped-PCPs’ recommendation behavior.

Furthermore, those PCPs who worked a few days a week (OR = 1.769, CI: 1.035–3.025), agreed with children’s perceived high susceptibility to pneumonia (OR = 2.185, CI: 1.074–4.445), the necessity of vaccination (OR = 1.802, CI: 1.009–3.219), and accepted the theory of herd immunity (OR = 1.717, CI: 1.101–2.677) were more likely to recommend the PCV13 vaccine. Likewise, those PCPs who agreed with parents of children with better family conditions were less likely to recommend PCV13 vaccination (OR = 0.813, CI: 0.677–0.977).

### 3.4. Reasons for Not Recommending the PCV13 Vaccine

A multiple-choice ranking question examined the main barriers for the two groups of HCWs to recommend the PCV13 vaccine through the frequency of the recommended behavior, respectively. For the pediatric care providers, Figure 1a shows that their primary concern was adverse reactions, however, as shown in Figure 1b, a majority of primary care providers cited the high price as the main barrier that discouraged them from recommending the PCV13 vaccine to parents. In the group of those who rarely recommended the PCV13 vaccine (*p* < 0.05), the greatest number of HCWs held the belief that they believed that recommending PCV13 vaccines was beyond their job responsibilities (41.09%) and ineffective (39.97%). Meanwhile, they showed more concern about the absence of an amicable environment to advise vaccination without a policy guarantee (50.32%).

## 4. Discussion

This study examined the recommendation practices for the PCV13 vaccine among Peds-PCPs and PCPs in China. The findings showed that primary care providers were more likely to recommend PCV13 than pediatric care providers. Specifically, PCPs recommended the vaccine to patients about 68.79% of the time, while Ped-PCPs recommended it 41.02% of the time. Further analysis revealed that healthcare workers in primary-level hospitals or community health centers were the most likely to recommend PCV13 in various situations. However, the social drivers that influence the recommendation practices of HCWs are complex. We identified areas of BeSD domains in forms of thinking and feeling, practical issues, social processes, and motivation to be positively associated with routine PCV13 recommendations by both Peds-PCPs and PCPs. Understanding the differences in vaccine hesitancy by different types of HCWs can guide targeted interventions to promote their recommendation practices, thereby increasing children’s vaccine coverage, especially for children who need special healthcare advice.

Consistent with existing global research, this study found that primary care providers were more likely to recommend vaccination, emphasizing their pivotal role in the healthcare system, especially in preventive care and routine immunizations [18]. This trend is evident in many other countries where primary care providers are essential in delivering comprehensive healthcare [19,20]. Compared to Ped-PCPs, the PCPs in this study were characterized by a lower prevalence of postgraduate degrees and a higher workload [21]. Our results suggest that PCPs with more recent and more comprehensive training in immunization protocols are more likely to recommend PCV13. This indicates the necessity for targeted educational programs to enhance the PCPs’ knowledge and confidence in vaccine recommendations. Given their important role in routine checkups, preventive care, and communication with parents, PCPs are fundamental in guiding immunization services. Their frequent and direct interactions with children’s caregivers enable them to disseminate information about vaccination, particularly non-NIP vaccines. These findings underscore the significant role of PCPs, who consistently showed a higher inclination to recommend PCV13 across various situations.

Compared to PCPs, a higher percentage of Ped-PCPs in this work held a master’s degree or higher and worked longer hours per day, but had less experience with vaccination-related tasks. Additionally, Ped-PCPs often held higher professional titles, which means that their role may have a greater acceptance and trust of the parents to influence their vaccination decisions. Factors such as recent vaccine training, knowledge of PCV13 risks, concerns that recommending PCV13 might be off-putting to parents, and understanding that the high price of PCV13 may worry parents significantly influenced the Ped-PCPs’ recommendation behaviors. Ensuring that Ped-PCPs are well-trained in PCV13 vaccination procedures and equipped with up-to-date knowledge and communication skills to effectively convey the vaccine’s benefits to caregivers is crucial to increase the likelihood of delivering appropriate vaccine recommendations to their patients. Additionally, training programs should educate Ped-PCPs about the patient’s perspective to better address their concerns and improve communication.

Using the four BeSD domains, the significant factors influencing both the Peds-PCPs’ and PCPs’ recommendation practices for PCV13 were categorized as thinking and feeling about vaccination, motivation to seek (or provide) vaccination, and social processes that drive or inhibit vaccination [11]. The providers’ familiarity with PCV13 and the perceived necessity of PCV13 reflected their vaccine confidence and perceived disease risk. In this context, trust in vaccination plays a critical role: providers who have a strong belief in the effectiveness and safety of the vaccine are more likely to convey confidence to the caregivers during daily work, thereby enhancing their recommendation behaviors. Additionally, the providers’ trust can influence their own attitudes toward the vaccine, shaping their perceived disease risk and vaccination benefits. Frequent consultations about PCV13 also require a proficient understanding of the vaccine’s benefits and safety. Results from this study showed that providers are more likely to recommend PCV13 when they are confident in the vaccine. Since PCV13 is only administered to children younger than 2 years, this study measured the physicians’ willingness to get the influenza vaccine, a non-NIP and self-paid vaccine against respiratory diseases, to reflect their awareness of preventing respiratory diseases through vaccination. The results show that providers willing to get the influenza vaccine are more likely to recommend PCV13. Additionally, workplace norms, such as a sense of responsibility to recommend pneumonia prevention measures, also influenced the providers’ recommendation practices.

To improve the recommendation practices for PCV13, it is essential to enhance the providers’ vaccine confidence and understanding of disease risk through ongoing education and training programs. While these programs should focus on the benefits and safety of PCV13, emphasizing real-world data and case studies, we recognize that education alone cannot address all of these challenges. Nevertheless, educational initiatives serve as feasible and essential interventions that can improve the providers’ vaccine confidence and knowledge. These education programs can be adapted to the specific contexts, making them applicable even in low-resource settings. Engaging healthcare workers in the design of educational courses and materials can also ensure that the content is relevant and practical, fostering a sense of ownership and increasing the likelihood of behavior change.

Increasing motivation can be achieved by fostering a culture that values preventive care. Additionally, addressing social processes, such as promoting a shared sense of responsibility for public health and normalizing vaccine discussions within healthcare teams, can create a supportive environment for vaccine advocacy. Non-financial incentives play a crucial role in this context. Introducing recognition programs and peer acknowledgment for high vaccination rates may also improve the providers’ motivation. By reinforcing the social norm of vaccine advocacy through non-financial incentives, healthcare providers may feel a greater sense of professional and social fulfillment in their roles.

Both Peds-PCPs and PCPs identified limited work capacity as a significant barrier to their recommendation behaviors regarding PCV13. This challenge is particularly pronounced among primary care providers, where those working fewer than 5 days per week were 1.77 times more likely to recommend PCV13 than those with greater workloads. When providers struggle to meet demands, whether in terms of quantity or quality, they may be required to take on additional responsibilities alongside vaccination, consultation, and registration. In such instances, the providers’ willingness to recommend PCV13, a voluntary vaccine, may be constrained due to concerns about increased workload burden.

The lack of policy support and national financing presents significant barriers to the physicians’ recommendation of the PCV13 vaccine in China. Without clear organizational guidance, physicians may exhibit hesitancy in supporting the vaccine, potentially impacting their willingness to recommend it to patients. Additionally, the absence of policy support and non-inclusion in the NIP may create a perception among healthcare providers and caregivers that PCV13 vaccination is not a priority or may not be necessary. Furthermore, the lack of national financing directly results in high vaccination costs for caregivers, totaling RMB 2000 yuan or USD 280 for full vaccination, which significantly influences the providers’ recommendation practices.

This study had several limitations. First, the recommendation practices of providers were self-reported, which might have introduced self-recall bias. Second, although we aimed for a representative sample by using a multistage sampling method, the survey results were not weighted based on demographic characteristics. Third, while the findings offer valuable national-level insights, their application to less developed rural areas necessitates careful consideration of the local factors, ensuring appropriate adaptation for those specific contexts. Fourth, the examination of the role of financial incentives on the providers’ recommendation behavior was only exploratory, and detailed information about the financial incentives and further in-depth analysis are required. Finally, this survey only included providers and not caregivers, whose perspectives are also important.

## 5. Conclusions

This national cross-sectional study highlights differences in the PCV13 recommendation behaviors between Ped-PCPs and PCPs in China. While HCWs in community health centers are more inclined to recommend PCV13, vaccine hesitancy persists among pediatric care providers. Given that children with weakened immunity often seek advice from pediatric care providers, these specialists have greater opportunities to influence vaccination decisions. To address this, tailored interventions are essential to reduce misconceptions and enhance confidence in vaccine safety and efficacy among different types of HCWs, particularly Peds-PCPs. Moreover, implementing sensible policies and practical actions to alleviate the workload and provide risk protection for HCWs can further support these tailored interventions, potentially leading to improved immunization practices in developing countries.

## Figures and Tables

**Figure 1 vaccines-12-01082-f001:**
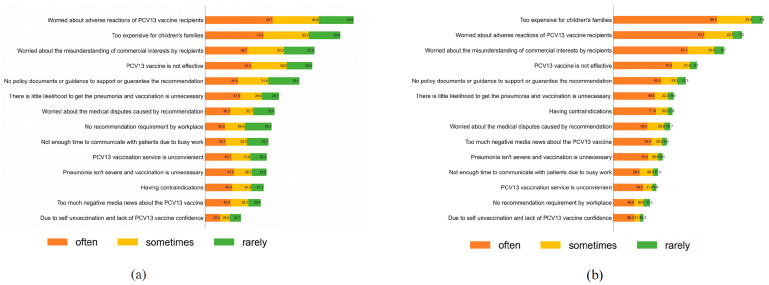
Impediments for pediatric care providers (**a**) and primary care providers (**b**) to introduce the PCV13 vaccine by recommended frequency (%). Note: Reasons for not recommending the PCV13 vaccine were not mutually exclusive. The percentage sum of all the reasons was more than 100%, as some HCWs chose more than one reason.

**Table 1 vaccines-12-01082-t001:** Characteristics of the surveyed pediatric care providers and primary care providers.

	Pediatric Care Providers N = 1775	Primary Care Providers N = 1804	*p*-Value
PALD, n (column%)			<0.001 *
Chongqing	316 (17.80)	246 (13.64)	
Jilin	395 (22.25)	596 (33.04)	
Shandong	310 (17.46)	285 (15.80)	
Shenzhen	433 (24.39)	402 (22.28)	
Beijing	321 (18.08)	275 (15.24)	
Gender, n (%)			<0.001 *
Male	502 (28.28)	305 (16.91)	
Female	1273 (71.72)	1499 (83.09)	
Age, n (%)			<0.001 *
<30	627 (35.32)	496 (27.49)	
31–40	623 (35.10)	673 (37.31)	
41–50	389 (21.91)	469 (26.00)	
>50	136 (7.66)	166 (9.20)	
Education level, n (%)			<0.001 *
Master’s degree and above	197 (11.10)	80 (4.43)	
Bachelor’s/Associate degree	1447 (81.52)	1487 (82.43)	
Senior secondary education	114 (6.42)	218 (12.08)	
Junior secondary education and below	17 (0.96)	19 (1.05)	
Hospital type, n (%)			<0.001 *
Primary hospitals/Community health centers	751 (42.31)	1216 (67.41)	
Secondary hospitals	313 (17.63)	247 (13.69)	
Tertiary hospitals	582 (32.79)	176 (9.76)	
Private hospitals	129 (7.27)	165 (9.15)	
Position, n (%)			<0.001 *
Executive	290 (16.34)	377 (20.90)	
Nonexecutive	1485 (83.66)	1427 (79.10)	
Vaccine training history in the past two years, n (%)			<0.001 *
Yes	1314 (74.03)	1644 (91.13)	
No	461 (25.97)	160 (8.87)	
Years of work experience within the organization, n (%)			0.009
≤5	899 (50.65)	820 (45.45)	
5–10	345 (19.44)	418 (23.17)	
10–20	320 (18.03)	328 (18.18)	
20–30	175 (9.86)	174 (9.65)	
>30	36 (2.03)	64 (3.55)	
Weekly work time, n (%)			<0.001 *
≤5	915 (51.55)	1243 (68.90)	
>5	860 (48.45)	561 (31.10)	
Average annual basic salary in CNY, n (%)			<0.001 *
≤50,000	566 (31.89)	842 (46.67)	
50,000–100,000	632 (35.61)	586 (32.48)	
100,000–200,000	397 (22.37)	306 (16.96)	
>200,000	180 (10.14)	70 (3.88)	
Income satisfaction, n (%)			<0.001 *
Yes	517 (29.13)	377 (20.90)	
No	1258 (70.87)	1427 (79.10)	
Familiarity with PCV13 vaccination procedures, n (%)			<0.001 *
Very familiar	274 (15.44)	748 (41.46)	
Relatively familiar	577 (32.51)	567 (31.43)	
Generally familiar	644 (36.28)	353 (19.57)	
Not very familiar	249 (14.03)	122 (6.76)	
Not familiar at all	31 (1.75)	14 (0.78)	
Frequency of vaccine consultation for parents, n (%)			<0.001 *
Never	170 (9.58)	381 (21.12)	
Hardly	448 (25.24)	723 (40.08)	
Sometimes	635 (35.77)	484 (26.83)	
Often	388 (21.86)	190 (10.53)	
Always	134 (7.55)	26 (1.44)	
Willingness to learn immune knowledge in the future, n (%)			<0.001 *
Very willing	868 (48.90)	1219 (67.57)	
Relatively willing	605 (34.08)	411 (22.78)	
Generally willing	276 (15.55)	153 (8.48)	
Not very willing	21 (1.18)	15 (0.83)	
Not willing at all	5 (0.28)	6 (0.33)	

Note: * *p*-value represents a comparison of the responses of the Ped-PCPs and PCPs. N total number of parents, n (%) number (percentage) of respondents in a given category, PLAD provincial-level administrative divisions, CNY, Chinese Yuan, CNY 1 = USD 0.14 on 1 April 2024.

**Table 2 vaccines-12-01082-t002:** Frequency of PCV13 recommendation among the two types of HCWs.

	Total (%)	Pediatric Care Providers (%)	Primary Care Providers (%)	*p*-Value
Recommendation rate of PCV13 vaccine				<0.001 *
Never	6.29	10.76	1.88	
Hardly	11.12	15.77	6.54	
Sometimes	27.58	32.45	22.78	
Often	32.72	27.10	38.25	
Always	22.30	13.92	30.54	
Mean score	3.54 ± 0.019	3.18 ± 0.028	3.89 ± 0.023	<0.001 *

The following formula was used to obtain the mean score:
$\frac{\sum_{k=1}^{5}k\times n_{k}}{\sum_{k=1}^{5}n_{k}}$, k is the frequency that HCWs recommend PCV13. * *p* < 0.05.

**Table 3 vaccines-12-01082-t003:** Frequency of PCV13 recommendation among HCWs from different levels of medical and sanitary institutions.

	Total (%)	Primary Hospitals/ Community Health Centers (%)	Secondary Hospitals (%)	Tertiary Hospitals (%)	Private Hospitals (%)	*p*-Value
Introducing information about the PCV13 vaccine during health education courses						
Recommendation rate	69.90	71.55	67.21	68.75	63.03	0.101
Introducing information about the PCV13 vaccine when parents actively consult about pneumonia-related information						
Recommendation rate	71.01	72.86	69.23	68.18	63.03	0.042
Introducing PCV13 vaccine information when children under 2 years old receive first-class vaccine						
Recommendation rate	69.68	72.29	63.16	66.48	63.64	0.006

**Table 4 vaccines-12-01082-t004:** Logistic regression to identify the influencing factors of PCV13 recommendation in pediatric care providers and primary care providers.

Factors	Pediatric Care Providers’ Recommendation Practice		Primary Care Providers’ Recommendation Practice	
	OR	95%CI	OR	95%CI
General characteristics				
Vaccine training history in the past two years				
Yes	1.470 *	(1.049–2.509)	1.128	(0.622–2.049)
No	ref			
Weekly work time (days)				
≤5	1.052	(0.748–1.479)	1.769 *	(1.035–3.025)
>5	ref			
Willing to get influenza vaccine in the future				
Yes	1.238 *	(1.050–1.458)	1.550 *	(1.180–2.036)
No	ref			
Average basic salary	1.245 *	(1.021–1.518)	1.649 *	(1.146–2.373)
Frequent inquiries about the vaccine	4.544 *	(3.721–5.547)	2.338 *	(1.850–2.956)
Overall PCV13-related knowledge				
Familiar with PCV13 vaccination procedures	1.277 *	(1.072–1.522)	1.692 *	(1.340–2.137)
Understanding that children have a healthy lifestyle does not mean that they will be free from pneumonia, and there is still a need to vaccinate against pneumonia	1.134	(0.639–2.011)	2.185 *	(1.074–4.445)
Knowing that the newly marketed PCV13 does not carry a higher risk than PCV7	1.986 *	(1.163–3.391)	1.569	(0.972–2.533)
Understanding that if a child is not vaccinated against pneumonia, it increases the risk of pneumonia for the entire population	1.222	(0.889–1.679)	1.717 *	(1.101–2.677)
Thinking and feeling				
Perceived high susceptibility to pneumonia and the necessity of vaccination	1.635 *	(1.115–2.397)	1.802 *	(1.009–3.219)
Practical issue				
Thinking that recommending PCV to parents would be undesirable	0.785 *	(0.663–0.929)	0.870	(0.686–1.104)
Thinking that the main reason why parents are reluctant to vaccinate their children against pneumonia is the high price	0.831 *	(0.707–0.977)	0.906	(0.723–1.134)
Agree that parents of children with better family conditions were more likely to accept the pneumonia vaccine recommendation	0.981	(0.861–1.118)	0.813 *	(0.677–0.977)
Social processes				
Agree that it is the responsibility of medical personnel to provide information and advice on pneumonia vaccinations to parents of children	1.353 *	(1.167–1.567)	1.442 *	(1.193–1.744)
Motivation				
Agree that their busy schedule has prevented them from introducing the pneumonia vaccine to the parents	0.750 *	(0.637–0.882)	0.645 *	(0.514–0.810)

* *p* < 0.05. OR: odds ratio. CI: confidence interval.

## Data Availability

The data presented in this study are available on request from the corresponding author. The data are not publicly available due to privacy reasons.

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
