# Peer review of "Differences in PCV13 Recommendation Practices between Pediatric Care Providers and Primary Care Providers in China: A Cross-Sectional Survey of Behavior and Social Drivers"

_vaccines, 2024, doi:10.3390/vaccines12091082_

Round 1

Reviewer 1 Report

Comments and Suggestions for Authors

This paper presents the findings from a study exploring the views of two differing classes of healthcare workers in China on the promotion of a specific vaccine, Pneumococcal Conjugate Vaccine. Using a self-completed questionnaire, the authors suggest that there is a difference between the two grades in the willingness to promote the vaccine and this is linked to a number of factors including vaccine cost. They conclude that “need for tailored interventions 30 to address differing perceptions and enhance confidence in the safety and efficacy of vaccines 31 among HCWs, particularly Peds-PCPs.”.

While the paper is of importance at present there are a number of shortcomings which preclude a recommendation to publish. These include:

The literature review should focus more on the wider aspects of shortcomings in the promotion of vaccines by health care workers in general. A key issue which the authors have not identified lies in trust. This has implications their conclusions, more about below. Two papers which are listed below would help provide a more nuanced and critical approach to the current work.

It is not clear from the present work exactly how the history of vaccination support in China has evolved. Why is the present vaccine apparently not as supported as other vaccines? This issue should be explored in the paper.

Allied to this is the issue of cost. How does the cost issue influence uptake? Are all vaccines cheaper? Who decides what the cost of vaccines are? What is the relative and absolute costs? Are some vaccines covered through cost-sharing while others not? If yes, why?

The platform Wenjuanxing is used. While the authors claim that any data are anonymised, who from the State might have access to any of the information. What does 20 RMB equate to? The authors write “..(t)hat the predicted PCV recommendation rate was 50%..). What does recommendation mean? 

The layout for Tables 1 and 2 is different with the column order being reversed. This is not helpful for comparisons to be made by the readers and it is suggested that the same order is adopted for both tables. 

The use of mean scores in Table 3 is not useful. By combining data derived from a qualitative scale, certain assumptions are being made and could/should be questioned.  Furthermore, it is not clear whether staff working in each of the differing institutions only work in those institutions. Can staff work in a number of them? 

The wording ‘repulsive’ (page 9, line 205) should perhaps be changed. 

On page 11 (lines 227-8) there is an interesting issue about job responsibilities. How does this square with other vaccination programmes. 

Lines 257 to 261 are non-sequiters. The rationale may be correct but the current work does not prove the relationship. There are a host of other factors (see comments above about other work in this field). Education programmes alone will not address other shortcomings including cost. Furthermore, knowledge alone does not necessarily lead to change. 

References.

Karafillakis E, Larson HJ. The paradox of vaccine hesitancy among healthcare professionals. Clin Microbiol Infect. 2018;24(8):799-800.

Kaur, M.; Coppeta, L.; Olesen, O.F. Vaccine Hesitancy among Healthcare Workers in Europe: A Systematic Review. Vaccines 2023, 11, 1657.

Author Response

Response to Reviewer 1 Comments

1. Summary

This paper presents the findings from a study exploring the views of two differing classes of healthcare workers in China on the promotion of a specific vaccine, the Pneumococcal Conjugate Vaccine. Using a self-completed questionnaire, the authors suggest that there is a difference between the two grades in the willingness to promote the vaccine and this is linked to a number of factors including vaccine cost. They conclude that “need for tailored interventions 30 to address differing perceptions and enhance confidence in the safety and efficacy of vaccines 31 among HCWs, particularly Peds-PCPs.”.

While the paper is of importance at present there are a number of shortcomings which preclude a recommendation to publish. These include:

Dear reviewer,

Thank you very much for taking the time to review this manuscript. We highly appreciate your efforts dedicated to providing feedback and valuable improvements to our manuscript. We have addressed all your valuable comments point-by-point. Please find the detailed responses below and the corresponding revisions highlighted in yellow in the re-submitted manuscript.

  1. Point-by-point response to Comments and Suggestions for Authors

Comments 1: The literature review should focus more on the wider aspects of shortcomings in the promotion of vaccines by health care workers in general. A key issue which the authors have not identified lies in trust. This has implications their conclusions, more about below. Two papers which are listed below would help provide a more nuanced and critical approach to the current work.

Response 1: Thanks a lot for your suggestion.

We agree that trust is strongly relevant to the context of this study. We have included two items in the survey to measure trust by HCWs in vaccine safety and effectiveness, i.e., "Knowing that the newly marketed PCV13 does not carry a higher risk than PCV7" and "Knowing that the newly marketed PCV13 does not carry a higher risk than PCV7" to measure the trust of Peds-PCP and PCP in vaccines.

We have edited to more clearly highlight the importance of trust in our manuscript.

In the Introduction, we have added: “As drivers of parents' vaccination decisions are multifaceted, vaccine hesitancy among healthcare workers (HCWs) play important roles in parents’ vaccination perceptions and decision-making. There have been increasing reports of vaccine hesitancy among healthcare workers in European countries, highlighting how their attitude can significantly influence parents’ decisions. This could be particularly relevant in China, as immunization information from HCWs in local hospitals and primary healthcare facilities is often considered the most trustworthy by many parents. However, previous studies have shown that 46.2% of public health workers in China were not willing to recommend any non-NIP vaccines to children, which is a barrier to optimizing non-NIP vaccine uptake. A critical element that contributes to vaccine hesitancy is trust. Trust between providers and caregivers form a cycle: where providers exhibit confidence in vaccines and communicate this effectively, it fosters trust among caregivers, which in turn encourages them to accept vaccination for their children. Conversely, if HCWs themselves express distrust or hesitancy about vaccines, it undermines caregiver trust, resulting in a counterproductive feedback loop that impede vaccination efforts.”  

In the Introduction section, we have also explained the limited engagement of health care workers for vaccine promotion, especially the Ped-PCPs.

“Pediatric care providers, on the other hand, work in children’s hospitals or general hospitals, often dealing with children who have medical conditions. Therefore, Ped-PCPs who are widely recognized by children's parents for their professional expertise, often garner a profound level of trust from these parents, despite not being directly responsible for administering vaccination services. Until now, few studies have been published on the recommendation practice for PCV13 of both PCPs and Ped-PCPs. Thus, the recommendations from both PCPs and Ped-PCPs are important for shaping parents' immunization decisions.”

In the Discussion Section, we further highlighted our analysis on how trust can affect providers’ vaccination recommendation practice. Besides, we elaborated the shortcomings in the promotion of vaccines by healthcare worker in general:

“In this context, trust in vaccination plays a critical role: providers who have a s strong belief in the effectiveness and safety of the vaccine were more likely to convey confidence to the caregivers during daily work, thereby enhancing their recommendation behaviors. Additionally, providers’ trust can influence their own attitudes towards the vaccine, shaping their perceived disease risk and vaccination benefits.”  

Comments 2: 

It is not clear from the present work exactly how the history of vaccination support in China has evolved. Why is the present vaccine apparently not as supported as other vaccines? This issue should be explored in the paper.

Response 2: We thank the reviewer for raising this important question. We have updated the Introduction Section with a brief overview of the history of vaccination support in China, clarified the differences between PCV13 and other NIP vaccines, and provided reasons for why PCV13 has yet to be included in the NIP:

“China’s National Immunization Program, funded by the central government, has been expanded several times since its establishment, which provides free vaccination to all entitled children for 14 vaccines covered under the program. Unfortunately, the 13-valent PCV (PCV13) has yet to be included in China’s NIP, making it a completely self-pay vaccine with no subsidies available. A variety of factors, such as disease burden, vaccine availability, vaccination capacity, cost-effectiveness, need to be considered when including a vaccine in the NIP. Consequently, the pooled vaccination rate of PCV13 in China is only 21.7% (95% CI: 17.2-26.5%), substantially lower than the global average and that of most Asian countries as reported by WHO Immunization data.”

Comments 3:

Allied to this is the issue of cost. How does the cost issue influence uptake? Are all vaccines cheaper? Who decides what the cost of vaccines are? What is the relative and absolute costs? Are some vaccines covered through cost-sharing while others not? If yes, why?

Response 3: Thank you for this comment. In our Introduction Section, we pointed out that PCV13 is a completely self-pay vaccine with no subsidies available:

“China’s National Immunization Program, funded by the central government, has been expanded several times since its establishment, which provides free vaccination to all entitled children for 14 vaccines covered under the program. Unfortunately, the 13-valent PCV (PCV13) has yet to be included in China’s NIP, making it a completely self-pay vaccine with no subsidies available. ”

Patients have to pay the market price (483-723 RMB or equivalent to 68-102 USD per dose), totaling RMB 2000-3500 yuan or USD 280-490 for a full PCV13 vaccination. There are no cost-sharing mechanisms for this vaccine.

In our Discussion Section, we also discussed how cost may influence providers’ recommendation practices:

“The lack of policy support and national financing presents significant barriers to physicians' recommendation of the PCV13 vaccine in China. Without clear organizational guidance, physicians may exhibit hesitancy in supporting the vaccine, potentially impacting their willingness to recommend it to patients. Additionally, the absence of policy support and non-inclusion in the NIP may create a perception among healthcare providers and caregivers that PCV13 vaccination is not a priority or may not be necessary. Furthermore, the lack of national financing directly results in high vaccination costs for caregivers, totaling RMB 2000-3500 yuan or USD 280-490 for full vaccination, which significantly influences providers’ recommendation practices.”

Comments 4:

The platform Wenjuanxing is used. While the authors claim that any data are anonymised, who from the State might have access to any of the information. What does 20 RMB equate to? The authors write “..(t)hat the predicted PCV recommendation rate was 50%..). What does recommendation mean? 

Response 4: Thank you for your careful review of our manuscript. We’ve modified the sentence to give the reimbursement amount in equivalent USD value:

“It takes about 10 minutes to complete the questionnaire, and each participant could receive a 20 RMB (equivalent to 2.80 USD) allowance for their time and support upon answering all questions.”

We would like to clarify that Wenjuanxing is an online survey tool designed specifically for conducting surveys and collecting data anonymously in China. The responses collected through this platform are anonymized, ensuring that individual participants cannot be identified from their responses. Furthermore, access to the anonymized data is restricted to authorized researchers only. While there may be regulatory authorities that have oversight over data collection processes, the design of Wenjuanxing prioritizes participant confidentiality and data security. We adhere to strict ethical guidelines to ensure that all collected information remains confidential and is used solely for research purposes. No identifiable personal information is linked to the survey responses, thus minimizing any potential for misuse of data by external entities.

We highly agree with your suggestion regarding the recommendation rate. To avoid misunderstanding, we’ve added:

“The initial sample size n required for the survey was calculated by the following formula based on an error of 0.05 and a maximum permissible error :=0.1p. p represents the predicted PCV13 recommendation rate of HCWs, referring to 50% (the most conservative estimated sample size).”

Comments 5:

The layout for Tables 1 and 2 is different with the column order being reversed. This is not helpful for comparisons to be made by the readers and it is suggested that the same order is adopted for both tables. 

Response 5: Thank you very much for pointing out this problem. All the column order has been unified with “pediatric care providers” on the left and “primary care providers” on the right.

Comments 6:

The use of mean scores in Table 3 is not useful. By combining data derived from a qualitative scale, certain assumptions are being made and could/should be questioned.

Furthermore, it is not clear whether staff working in each of the differing institutions only work in those institutions. Can staff work in a number of them? 

Response 6:

We appreciate your feedback regarding Table 3. In response to your concerns, we have revised Table 3 to present only the frequency of providers recommending the vaccine across different levels of medical and sanitary institutions. No qualitative scales were used in our analysis. Instead, each provider was asked a specific question regarding whether they would recommend the vaccine in each scenario presented.

Moreover, healthcare providers are only allowed to work full-time in one facility, as required by the national law. During data collection, each response had an unique survey ID according to their social media account, allowing us to prevent duplicate samples while keeping the data deidentified.

Comments 7:

The wording ‘repulsive’ (page 9, line 205) should perhaps be changed. 

On page 11 (lines 227-8) there is an interesting issue about job responsibilities. How does this square with other vaccination programmes. 

Response 7: We strongly agree with your suggestion and have changed the wording to “undesirable”

We added clarification regarding your second comment in the Introduction:

“Healthcare workers are not required to recommend non-NIP vaccines as part of their official duties, such recommendations are voluntary. However, the willingness of different types of healthcare workers in China to recommend PCV13 remains unclear.”

Recommending NIP vaccines is an obligation of healthcare workers in China.

Comments 8:

Lines 257 to 261 are non-sequiters. The rationale may be correct but the current work does not prove the relationship. There are a host of other factors (see comments above about other work in this field). Education programmes alone will not address other shortcomings including cost. Furthermore, knowledge alone does not necessarily lead to change. 

Response 8:We thank the reviewer for this comment. We mentioned targeted educational programs based on our finding that PCPs who received frequent vaccination training had higher recommendation rates. We may have previously focused too much on training for medical staff, but we believe that this factor is also the easiest to intervene on and because of the uneven distribution of healthcare resources in China, education programs can also be adapted and applied to rural areas. Therefore, we have made the following changes in the Discussion Section:

“To improve recommendation practices for PCV13, it is essential to enhance providers' vaccine confidence and understanding of disease risk through ongoing education and training programs. While these programs should focus on the benefits and safety of PCV13, emphasizing real-world data and case studies, we recognize that education alone cannot address all challenges. Nevertheless, educational initiatives serve as a feasible and essential intervention that can improve providers’ vaccine confidence and knowledge. These education programs can be adapted to the specific contexts, making them applicable even in low-resource settings. Engaging healthcare workers in the design of educational courses and materials can also ensure the content is relevant and practical, fostering a sense of ownership, and increasing the likelihood of behavior change.”

Reviewer 2 Report

Comments and Suggestions for Authors

Dear authors,

I have now completed the review of the manuscript titled "Differences in PCV13 Recommendation Practices between Pediatric Care Providers and Primary Care Providers in China: A Cross-Sectional Survey of Behavior and Social Drivers."

The manuscript is interesting and, in general, fairly well-written.

I have some suggestions to further improve the quality of the manuscript.

I would like to suggest that the authors address these limitations in the article, either by discussing them in the limitations section or, where feasible, by making the appropriate revisions:

1. The introduction should be expanded. Provide readers more insights of vaccine acceptance factors, including another Asian country, which may have similarities or contrasts with the Chinese context. For example, acceptance of COVID-19 vaccines in South Korea and Japan would be helpful. Especially, national prevalence and socioeconomic factors associated with the area.

2. Some methodologies are seems vague to me. For example, unclear rationale for the choice of covariates included in the logistic regression models. Also, potential for multicollinearity between some predictor variables (e.g. familiarity with procedures and recent training) not addressed. Lastly, limited information on the validation process for the survey instrument.

3. Findings may not be fully generalizable to rural areas or smaller healthcare facilities not included in the sample. And the study does not account for regional variations in healthcare policies or resources that could impact recommendation practices.

4. Discussion of how the findings compare to similar studies in other countries or healthcare systems is limited. Also, the impact of the COVID-19 pandemic on vaccination attitudes and practices is not explicitly addressed.

Thank you for your valuable contributions to our field of research. I look forward to receiving the revised manuscript.

Author Response

Response to Reviewer 2 Comments

1. Summary

I have now completed the review of the manuscript titled "Differences in PCV13 Recommendation Practices between Pediatric Care Providers and Primary Care Providers in China: A Cross-Sectional Survey of Behavior and Social Drivers."

 The manuscript is interesting and, in general, fairly well-written.

 I have some suggestions to further improve the quality of the manuscript.

I would like to suggest that the authors address these limitations in the article, either by discussing them in the limitations section or, where feasible, by making the appropriate revisions:

Dear reviewer,

Thank you very much for taking the time to review this manuscript. We highly appreciate your efforts dedicated to providing feedback and valuable improvements to our manuscript. Please find our responses to your comments below, the corresponding revision is highlighted in yellow in the re-submitted manuscript.

  1. Point-by-point response to Comments and Suggestions for Authors

Comments 1:

  1. The introduction should be expanded. Provide readers more insights of vaccine acceptance factors, including another Asian country, which may have similarities or contrasts with the Chinese context. For example, acceptance of COVID-19 vaccines in South Korea and Japan would be helpful. Especially, national prevalence and socioeconomic factors associated with the area.

Response 1:  Thank you for your valuable feedback and suggestions for expanding the introduction section of our manuscript. We appreciate your guidance in providing readers with a more comprehensive understanding of vaccine acceptance factors, particularly in the context of Asian countries. We have revised the introduction section to include additional insights on PCV13 vaccination rates in the West Asian Pacific region, recognizing the importance of regional comparisons. This information will help contextualize our findings within a broader global perspective.

Furthermore, we have emphasized the growing phenomenon of vaccine hesitancy among healthcare professionals, particularly among pediatricians, who play a crucial role in influencing parental decisions regarding childhood vaccinations. Recognizing the scarcity of systematic reviews on vaccine trust among healthcare workers in low- and middle-income countries (LMICs), we highlight the significance of our study in contributing to this evidence base. Our research aims to fill this gap by providing insights into the PCV13 recommendation practices among pediatric care providers and primary care providers in China, an LMIC setting.

To be specific, in the Introduction, we have added:

Consequently, while the pooled vaccination rate of PCV13 in China is only 21.7% (95% CI: 17.2-26.5%), substantially lower than the global average and that of most Asian countries as reported by WHO Immunization data. Within the Western Pacific Region of the WHO, the PCV13 coverage is 26%, while European Region boast significantly higher vaccination rates of 86% for PCV13, underscoring a regional disparity. The gap between NIP vaccine coverage and PCV13 coverage highlights a significant area for improvement.

Recent systematic review has highlighted a gap in research on vaccine hesitancy and trust in vaccination from the perspective of healthcare providers, particularly in low- and middle-income countries. Our research aims to address this gap by identifying factors that affect providers’ vaccine recommendations. Specifically, this study identifies changeable factors across four BeSD domains that hinder healthcare providers’ recommendation for PCV13. By understanding these barriers, we can design and promote more targeted interventions to change provider behavior and improve PCV13 coverage in a wider population.

.Comments 2:

  1. Some methodologies are seems vague to me. For example, unclear rationale for the choice of covariates included in the logistic regression models. Also, potential for multicollinearity between some predictor variables (e.g. familiarity with procedures and recent training) not addressed. Lastly, limited information on the validation process for the survey instrument.

Response 2: 

Thanks a lot for your suggestion.

We acknowledge your point about the unclear rationale for the selection of covariates in our logistic regression models. In the section of 2.3 Statistical analysis, we have included a more detailed description of our variable selection process:

“For univariate analysis, the chi-square test and Wilcoxon rank-sum test were performed to assess the association between PCV13 recommendations with each of the independent variables. Independent variables tested to be significant were then further entered into the multivariate logistic regression model where odds ratios (OR) and 95% confidence intervals (CI) were calculated to examine the influencing factors of recommendation behaviors of healthcare workers. ”

To investigate the possibility of multicollinearity between predictor variables, we utilized Variance Inflation Factors (VIFs) and tolerances. We considered VIF values less than 5 and tolerance values greater than 0.1 as indicators of no significant collinearity. As detailed in the attached supplementary Tabel 1 (Appendix 2), the VIF values for all predictor variables, including familiarity with procedures and recent training, are well below the threshold, indicating that multicollinearity is not a significant concern in our model. Specifically, it is likely that recent training focused on general vaccination practices and updates in the National Immunization Program, rather than delving into the specifics of non-NIP vaccine protocols like PCV13. And the accuracy and stability of the models were evaluated using the Hosmer-Lemeshow goodness-of-fit test.

In the 2.3 Statistical analysis and 3.2. Attitudes and behaviors for recommending PCV13 section, we further highlighted our analysis on potential multicollinearity:

“Variance inflation factors(VIFs) and tolerances were used to investigate for variable collinearity, with VIF less than 5 and tolerance greater than 0.1 considered to indicate no significant collinearity. The accuracy and stability of the models were evaluated using the Hosmer-Lemeshow goodness-of-fit test. ”

“The collinearity diagnostic analysis showed that the VIFs of those risk factors were less than 4, suggesting that there is no strong indication of multicollinearity among variables (Appendix.2 Supplementary Table 1). The results of Hosmer-Lemeshow test indicated that indicated that both the Ped-PCP's ( = 9.569, P= 0.297) model and the and the PCP's model ( = 6.492, P = 0.592) exhibited good overall fits. ”

In response to your concern about limited information on the validation process for our survey instrument, we have provided additional details in the Materials and Methods section, specifically under the 2.2 Measures subsection:

The guidelines published by Department of Immunization, Vaccines and Biologicals (IVB),WHO provided knowledge basis for the questionnaire. The preliminary questionnaire underwent testing via face-to-face interviews with pre-selected participants. Initially, respondents completed the questionnaires independently. Subsequently, they were interviewed about their comprehension of questions and any challenges encountered during completion. Based on their feedback, adjustments were made to content and phrasing, finalizing the questionnaires.

Comments 3:

  1. Findings may not be fully generalizable to rural areas or smaller healthcare facilities not included in the sample. And the study does not account for regional variations in healthcare policies or resources that could impact recommendation practices.

Response 3: Thanks for raising your concern regarding the generalizability of our findings. To address this, we would like to provide an overview of our sampling method.

In this survey, we employed a multistage sampling approach. Initially, we selected five pilot areas for local development (PALDs) — Chongqing, Jilin, Shandong, Shenzhen, and Beijing — based on China’s Division of Central and Local Financial Governance and Expenditure Responsibilities in the Healthcare Sector, which stratifies the 31 PALDs into five layers. Following that, medical institutions within each province or municipality were categorized into three levels based on the region's economic development level (GDP), which were classified as low, medium, and high. Our study ensured that the minimum sample size across these three levels was satisfied, resulting in a national sample that encompasses regions with varying economic statuses.

The distribution of participants in our study reflects the general composition of healthcare providers in China, with 42% of Peds-PCPs and 67% of PCPs coming from community and primary health centers. This is significant as community and primary health centers are more prevalent in less developed regions, whereas tertiary hospitals tend to be concentrated in larger cities.

Additionally, we acknowledge that there may be regional variations in healthcare policies and resources that could impact recommendation practices. However, our suggestion to enhance practitioner education is designed to be an adaptable intervention that can be tailored to different contexts based on the local environment. This flexibility allows for the incorporation of specific regional factors to improve vaccine recommendation practices effectively.

In the paragraph discussing limitations, we have added:

Third, while the findings offer valuable national-level insights, their application to less developed rural areas necessitates careful consideration of local factors, ensuring appropriate adaptation for those specific contexts.

Comments 4:

  1. Discussion of how the findings compare to similar studies in other countries or healthcare systems is limited. Also, the impact of the COVID-19 pandemic on vaccination attitudes and practices is not explicitly addressed.

Response 4: We thank the reviewer for this insightful comment. We acknowledge that the COVID-19 pandemic has significant impact, particularly in shaping providers' attitudes towards vaccination.

In the post-COVID context, we have observed a notable lack of research specifically addressing this topic in similar settings, as we note in our Introduction:

Recent systematic review has highlighted a gap in research on vaccine hesitancy and trust in vaccination from the perspective of healthcare providers, particularly in low- and middle-income countries. Our research aims to address this gap by identifying factors that affect providers’ vaccine recommendations, which is critical for reducing disease burden in China. Specifically, this study identifies changeable factors across four BeSD domains that hinder healthcare providers’ recommendation for PCV13. By understanding these barriers, we can design and promote more targeted interventions to change provider behavior and improve PCV13 coverage in a wider population.”

The pandemic has had complex effects on healthcare providers' attitudes toward vaccinations, with some exhibiting increased confidence due to the heightened emphasis on public health measures, while others may have developed hesitancy or skepticism. In light of these observations, we propose enhanced educational initiatives as a critical response to our findings. By providing targeted training and resources, we aim to address any negative shifts in attitudes and reinforce positive perspectives on vaccination among healthcare providers. This approach can help improve vaccination recommendation practices moving forward.

Reviewer 3 Report

Comments and Suggestions for Authors

The manuscript entitled "Differences in PCV13 recommendation practices between pediatric care providers and primary care providers in China: a cross-sectional survey of behavior and social drivers” by Dang et al. is not suitable for publication in Vaccines Journal. The quality of the manuscript is poor. The data analysis and presentation are also very poor. All the results of the study are reported in the form of Tables excluding a figure. The results are also not discussed adequately, and data analysis has great scope for improvement. Hence, I recommend this manuscript for rejection.

Author Response

Response to Reviewer 3 Comments

Summary

The manuscript entitled "Differences in PCV13 recommendation practices between pediatric care providers and primary care providers in China: a cross-sectional survey of behavior and social drivers” by Dang et al. is not suitable for publication in Vaccines Journal. The quality of the manuscript is poor. The data analysis and presentation are also very poor. All the results of the study are reported in the form of Tables excluding a figure. The results are also not discussed adequately, and data analysis has great scope for improvement. Hence, I recommend this manuscript for rejection.

ResponseThank you for taking your valuable time to review our manuscript. We appreciate the constructive feedback you have provided, and tried our best to revise our manuscript based on your comments and those of the other reviewers. We have engaged a native English speaker to refine the language, ensuring clarity and fluency throughout the text.

To address the concerns regarding data analysis and presentation, we have undertaken a series of enhancements to our methodological approach. Specifically, we have included a more nuanced and thorough description of our variable selection process in the revised manuscript. As detailed in the attached supplementary Tabel 1 (Appendix 2), we utilized Variance Inflation Factors (VIFs) and tolerances as diagnostic tools to assess the degree of collinearity in our model to investigate the potential for multicollinearity between variables. To validate the accuracy and stability of our models, we have performed the Hosmer-Lemeshow goodness-of-fit test. Regarding the validation process for our survey instrument, we have expanded upon the details provided in the original manuscript.

Furthermore, we have enhanced the clarity and quality of the figures and tables in the results section, ensuring that they visually illustrate key findings in a more accessible and interpretable manner. We have updated these figures in Appendix 1 and included them, along with the revised text and tables, in the manuscript.

We have also rewritten the discussion section to provide a more thorough examination of our results. We have delved deeper into the implications of our findings, contextualizing them within the broader literature and discussing potential explanations for the observed differences in PCV13 recommendation practices between pediatric and primary care providers.

We believe these revisions have significantly improved the manuscript in terms of both content and presentation. We really hope you can reconsider your initial assessment and re-evaluate the updated version of our work. We are confident that the enhancements made will meet the standards expected for publication in Vaccines Journal. Thank you again for your feedback.

Round 2

Reviewer 1 Report

Comments and Suggestions for Authors

One very minor issue. Reference 6 in the body of the text is not superscript. That part the present paper is a substantial improvement on the first version and would add to the literature in this field.

Reviewer 2 Report

Comments and Suggestions for Authors

All comments were addressed.

Reviewer 3 Report

Comments and Suggestions for Authors

author thoroughly revised the manuscript and addressed the comments. hence recommended for accept for publication